# Unveiling the Hidden Causes: Identifying the Drivers of Human–Elephant Conflict in Nilgiri Biosphere Reserve, Western Ghats, Southern India

**DOI:** 10.3390/ani14223193

**Published:** 2024-11-07

**Authors:** Nagarajan Baskaran, Selvarasu Sathishkumar, Varadharajan Vanitha, Mani Arjun, Perumal Keerthi, Nikshepan Goud Bandhala

**Affiliations:** 1Mammalian Biology Lab, Department of Zoology & Wildlife Biology, A.V.C. College (Autonomous) Mannampandal, Mayiladuthurai 609 305, India; ksathish605@gmail.com (S.S.); arjunm1430@gmail.com (M.A.); pkeerthi015@gmail.com (P.K.); nikshepangoudbandhala2222@gmail.com (N.G.B.); 2Department of Zoology, Dharmapuram Gnanambigai Government Arts College for Women, Mayiladuthurai 609 001, India; vanithadggc@gmail.com

**Keywords:** crop damage, ecological covariate, elephant electrocution, grass biomass, human casualties, property damage

## Abstract

The increasing human population and its resultant demand on land bring more people in contact with wildlife, leading to increasing human–wildlife conflict. The Asian elephant, a wide-ranging species, is being increasingly threatened by human–elephant conflict [HEC]. Its conservation is thus demands for identifying the causes of HEC and implement measures to mitigate the HEC effectively. To identify the causes of HEC, we assessed the HEC in five forest divisions in Western Ghats, India, and compared it with 26 ecological factors. Spatially, HEC was highest in the Territorial Forest Division and lowest in those declared as Protected Areas earlier. Further, our results showed that crop damage decreased with increasing grass biomass, elephant density, dry-thorn and deciduous habitats, and forest range area, but increased with adult males and forest range perimeter. Similarly, the property damage by elephants increased with crop damage frequency and human settlement/cultivation area, but decreased with grass biomass, forest range area, and deciduous habitat area. Human casualties due to elephants increased with property damage, temperature, and forest range perimeter, but decreased with grass biomass. These findings suggest that anthropogenic pressure that decreases the grass biomass and degrades the habitat is the likely root cause of HEC. Therefore, minimizing anthropogenic pressure would reduce HEC and ensure the long-term conservation of elephants and other species, which will ultimately enhance the economic growth of humans.

## 1. Introduction

The current global human population of 8.1 billion is expected to exceed 10 billion by the middle of this century [1], and the shifting patterns of global wealth and economic growth at the expense of ecological integrity has imposed a tremendous burden on the planet’s climate, biodiversity, and its life-supporting capacity [2]. With the current scenario of enormous changes to environmental health, understanding how organisms are coping with the changing environment is crucial to formulating strategies for managing the natural environment along with its biodiversity without affecting its life-supporting functions and allowing humans to live in harmony with nature. Given their large body mass and higher metabolic requirements, large mammals are increasingly facing anthropogenic pressure, leading to a decline in their populations and increasing human–large mammal conflict throughout the world [3]. Wide-ranging umbrella species like the Asian elephant, an endangered species [4], are increasingly coming into conflict with humans [5] due to the large-scale transformation of their natural habitats to meet the growing demands of the increasing human population [6]. Therefore, the conservation of umbrella species in their natural habitats is becoming an uphill task throughout Asia [7]. The global effort to conserve Asian elephants largely depends on identifying the drivers of human–elephant conflicts (HEC) and implementing appropriate measures to mitigate HEC. The consequences of HEC take many forms, from the death and disability of humans and elephants to the loss of crops, livestock, water resources, infrastructure, and even the emergence of communities that are unable to live without a fear of other communities [8].

Factors like the loss, fragmentation, and degradation of habitats, proliferation of alien invasive species, and increases in elephant populations have been identified as the prime factors driving conflict [7,9,10]. HEC seems to be greater in and around natural habitats with a higher degree of loss, degradation, and fragmentation. Elephant habitat size and its structural dimensions are also reported to play a vital role in HEC [5,11]. Changes in land use and landcover (LULC) elements, especially the conversion of private and revenue forests into human settlements/agricultural areas, are expected to increase HEC [5,12] as elephant clans show strong fidelity to their home and seasonal corridor [12]. Further, cultivated crops are more nutritive than wild forage so, as an optimal foraging strategy, adult bulls raid crop fields [5]. Intensive anthropogenic pressure in elephant habitats, especially in the form of over-grazing by cattle and for firewood and dung collection, is not only known for reducing edible fodder resources for herbivores, but also for promoting the proliferation of unpalatable and alien invasive plant species [13]. The specific driver(s) influencing HEC could vary spatially for each population [11]. Therefore, identifying the drivers of HEC and suggesting appropriate measures to mitigate HEC are vital for developing site-specific conservation plans for Asian elephants.

Due to the declining digestive efficiency, the dietary requirement of large herbivores increases allometrically with increasing body size [14]. Therefore, food abundance, a key predictor of animal species abundance [15,16], is especially important in shaping the ecology of large herbivores. Thus, assessments of food abundance are vital for understanding the ecological relationship of large herbivores with their habitats [17]. Understanding the abundance of grass species that form the principal diet of Asian elephants [18] would provide a better understanding of HEC, as it plays a major role in the ecology of elephants [14,19]. Despite the significant role of grass abundance, many of the studies that identified the causes of HEC have not recognized the importance of grass abundance and related it to HEC in both Asia ([20]—Sri Lanka, [21]—Southern India, [22]—Northeastern India, [23]—Bangladesh, [24]—Northeastern India, [25]—Southern India) and in Africa ([26]—Tanzania, [27]—Botswana). This is partly due to the fact that large herbivores are wide ranging and thus any field-based assessment of grass abundance is laborious. To overcome this, a few studies on HEC in Asia ([28]—Tamil Nadu, [29]—Sri Lanka, [30]—Myanmar, [31]—Nepal) used the Normalized Difference Vegetation Index (NDVI), a satellite-based vegetation index, as an indirect measure of forage abundance. However, ref. [32] examined the use of the NDVI as a proxy for food abundance for herbivores and demonstrated that the NDVI is not a good predictor of elephant forage in tropical forests, as the NDVI is negatively related to the total abundance of graminoids, which represent a bulk of the diets of elephants and other herbivores, and is positively related to canopy cover.

While studies on HEC have increased over the years, none of them related the field-based grass biomass covariate with HEC, a prime factor that decreases significantly with anthropogenic pressure and the proliferation of alien invasive species over the years in tropical forests. This study differs from the existing studies, as it compared all ecological covariates starting from weather, habitat, grass dynamics, and elephant population size and structure to anthropogenic pressure with HEC, and comprehensively identified the drivers of crop and property damage, human casualties, and elephant mortality due to conflict. The study was carried out covering five forest divisions, which experience a gradient of HEC, in the Nilgiri Biosphere Reserve, Western Ghats, which supports the largest global population of Asian elephants.

## 2. Methodology

### 2.1. Study Area

The study took place in the Nilgiri Biosphere Reserve (NBR), which is located at 12°15′–10°45′ N, 76°0′–77°15′ E, and is spread over an area of 5520 km^2^. The reserve is situated at the junction of three southern states—Tamil Nadu, Karnataka, and Kerala. It has an undulating terrain with an average elevation of 1000 m above msl. Rivers such as Nugu, Moyar, and Bhavani, as well as most of their tributaries, which are perennial and drain the area, provide water sources to wild animals throughout the year. Its varied reliefs and topography provide a diverse climate, with a short dry season (January to April) and long wet seasons from the southwest (May to August) and northeast (September to December) monsoons. The mean annual rainfall varies from 600 on the eastern side to 2000 mm on the western side. Corresponding to the gradient in rainfall, the vegetation varies from southern tropical dry-thorn forest on the east to semi-evergreen and moist deciduous forest on the west, with dry deciduous forest in between the two [33]. The NBR, along with its adjoining natural habitats, has remarkable faunal diversity and is well-known for supporting the largest population of Asian elephants, with an estimated population of 5750 individuals [34]. Deciduous forests of this area support a high biomass of grass ([18]—Appendix A). Overgrazing by domestic cattle and due to firewood collection are serious problems on its eastern fringes [13]. The landscape is identified as one of the important elephant ranges at the continent scale, with ongoing developmental activities [35]. Therefore, this study is essential for understanding the effect of developmental activities on HEC. Considering the prevalence of HEC in the NBR, the study covered three different sites (Figure 1), viz. the (i) Coimbatore Forest Division (CFD), (ii) Sathiyamangalam Tiger Reserve (STR), and (iii) Mudumalai Tiger Reserve (MTR) from Tamil Nadu that form a part of the NBR. For administrative purposes, both tiger reserves are further divided into two forest divisions each: the Sathiyamangalam and Hasanur Forest Divisions for the STR and the Mudumalai and Masinagudi Forest Divisions for the MTR, with the former division in each area (Hasanur and Mudumalai) as the core area and the latter (Sathiyamangalam and Masinagudi) as the buffer. Similarly, these five forest divisions are further divided into 25 forest ranges. Due to the present and past conservation statuses, HECs are expected to vary among these divisions. Likewise, within the tiger reserves, the core areas are expected to experience less anthropogenic pressure due to a lower level of human settlement/cultivation compared to the buffer areas.

### 2.2. Assessing the Intensity and Drivers of Human–Elephant Conflict

Secondary data on the intensity of HEC were collected from the forest department sources of the study area for the last six years, from 2018 to 2023. Ex-gratia applications submitted by the affected people to the forest ranges belonging to five forest divisions were examined and the number of farmers/people who experienced (i) crop damage, (ii) property damage, (iii) casualties due to elephants, and (iv) the number of elephants electrocuted in each forest range annually were recorded. In total, during the six-year period, the study area experienced damage to 4073 farmers’ crop fields, reflecting the intensity of crop damage, 220 cases of property damage, and 231 cases of human casualties due to elephants, and, as a consequence, 23 elephants died due to electrocution.

### 2.3. Ecological Covariate Assessment

The study overlayed 25 km^2^ grids on the study area map (Figure 1), selected alternating grid cells, and employed a systematic grid sampling method to collect ecological covariates. Additional layers, viz. boundaries of the study site, forest division, and forest ranges, and land use and landcover (LULC) elements, were added to the study area map. In total, 26 ecological covariates were assessed and these included weather variables, viz. (1) rainfall, (2) temperature, and (3) humidity; grass dynamics variables, viz. (4) grass height, (5) grass cover, (6) grass biomass index, (7) soft-texture grass, (8) hard-texture grass, (9) green grass, (10) dry grass, and (11) grass reproductive phase (grass flower and seeds); extent of various LULC elements, viz. (12) tropical evergreen/semi-evergreen habitats, (13) tropical deciduous habitats, (14) tropical dry-thorn or scrub habitat, (15) human settlement/cultivation area, and (16) water bodies; elephant population and habitat variables, viz. (17) population size, (18) elephant density, (19) adult male %, (20) area of forest range, (21) forest range, and (22) forest range perimeter; and anthropogenic pressure covariates from (23) cattle grazing, (24) firewood collection, (25) disturbance by locals and tourists, and (26) study period (2018–2023).

Monthly data on weather parameters, viz. rainfall, ambient temperature, and humidity, were obtained once for the center of each grid cell from an online source and averaged annually for 2018–2023. Data on grass dynamics parameters were collected by placing 1 km lines transecting in each 25 m^2^ grid cell, forming eight 1 m^2^ quadrats, and the eight grass parameters were assessed, which are listed and described in Table 1, at bimonthly intervals. The data collected were averaged annually for each forest range by considering the grids within each boundary. The LULC elements of the study area were obtained from an online source, as shown in Table 1. The elephant population parameters available at the forest division scale from the synchronized elephant censuses in 2017 and 2023 were considered in this study.

### 2.4. Statistical Analysis

The data on the dependent factors, viz. crop damage, property damage, human casualties due to elephants, and elephant electrocutions were entered into an Excel sheet and used to create an HEC database for each year for the 25 forest ranges. The collected ecological covariates were also compiled for each range and were included in the HEC database. To compare the dependent factors with the 25 covariates and identify the potential drivers of HEC, we used the Generalized Linear Mixed Model (GLMM) in the *lme4* package (using the glmer function) of R in the R program, version 4.3.2, as the dependent factors experienced pseudo-replication in multiple years. The dependent variables had multiple zeros and hence, they were adjusted by adding a constant of 1 using the Laplace smoothing technique to remove zeros before performing the analysis. Further, prior to testing using the GLMM, the Shapiro normality test was used to test the normality of the response variables. As the response variables were not normal, we explored log, square root, inverse, rank, and box transformations (using the boxcox function in the *MASS* package in R). However, none of the response variables could be normalized, so we incorporated Poisson distributions in the GLMM, as our response variables were continuous and slightly positively skewed. We checked skewness using the *e1071* package (with a skewness function) in R. Further, to identify collinearity among the covariates, we ran the Farrar–Glauber Collinearity test (using the ols_vif_tol function) in the *olsrr* package in the R program and based on VIF values, covariates with a VIF value < 10 were retained in the GLMM equation and those with a value > 10 were excluded [36,37]. In the GLMM equation, all the covariates, except for the forest range and year, were treated as the fixed effect covariates, while the forest range and year were treated as the random effect covariates. Following the same procedure used for comparing the dependent factor of crop damage with ecological covariates, we tested the correlation between property damage and human casualties with the ecological covariates separately in the GLMM. In addition to the ecological covariates listed in Table 1, we included the frequency of crop damage into the GLMM equation as a covariate for property damage; in addition, the property damage was used as a covariate for human casualties and elephant electrocution cases, as elephants that raid crop fields also damage properties, viz. houses and shops, to access the crop grains, and salt, borewell pipe lines, etc., to access water and cause human casualties deliberately when the victim tries to prevent them from damaging the crop or accidentally while damaging the house. Further, the crop-raiding elephant, when entering the crop field, also gets electrocuted by illegal electric fences, which are hooked to high-voltage lines. Illegal incidents took place in the study area several times.

## 3. Results

### 3.1. Overall HEC

Among the four different consequences of HEC, crop damage was the most frequent event (679 ± 83/annum), followed by human casualties (39 ± 4.1), property damage (37 ± 6.8), and elephant electrocution cases (4 ± 0.60). Annually, the three park managements together paid USD 167,810 ± 40,095 towards HEC, with human casualties drawing the bulk (USD 95,723 ± 24,027), followed by crop damage (USD 70,585 ± 16,930) and property damage (USD 1503 ± 454) by elephants.

### 3.2. Spatial Variation in HEC

Among the five forest divisions, the Coimbatore Division experienced significantly higher crop and property damage and human casualties, followed by the Forest Divisions of the STR and the crop and property damage and human casualties were lowest in the MTR (Figure 2). However, in electrocution cases, the core area of the Hasanur Forest Division had the highest number, followed by Coimbatore and Sathiyamangalam, with an equal number and the lowest number in the Masinagudi and Mudumalai Divisions.

### 3.3. Temporal Variation in HEC

Among the four different consequences of HEC assessed in the study, only crop damage (Figure 3A) increased significantly over the years. The property damage and human casualties numerically showed an increasing trend, while elephant electrocution showed a declining trend over the years, but they were not statistically significant (Figure 3).

### 3.4. HEC in Relation to Ecological Covariates—Univariate Analysis

The effect of each covariate on HEC revealed significant associations between the covariates and human–elephant conflict (Appendix A). For instance, crop damage was significantly higher in areas with less soft-texture grass compared to those with more soft-texture grass and the trend was same with the grass biomass index and elephant density. In contrast, crop damage was higher in areas with a higher adult male % in the population compared to areas with a lower adult male %. Similarly, property damage and human casualties were more prevalent in areas with more crop damage incidents and higher percentages of adult male elephants, but less prevalent in areas with a higher grass biomass index. On the other hand, elephant mortality by electrocution was more likely in regions with a higher elephant density and greater dry-thorn and deciduous habitat coverage, and lower in higher semi-evergreen coverage areas.

### 3.5. Drivers of HEC—Multivariate Analysis

**Crop damage:** A comparison of the ecological covariates with 4073 crop damage incidents in a multivariate framework showed that the incidents of crop damage decreased significantly in forest ranges with a higher grass biomass index, area of forest range, elephant density, green grass, and extent of dry-thorn and deciduous habitats, but increased with the area of forest division, percentage of adult males, and perimeter of forest range (Table 2). The significant contribution by 10 covariates, including the forest range and year under the random effect covariates, along with insignificant contributions by 6 other covariates, explained 97% of the variation in crop damage incidents. The finding that year and forest range were the significant predictors under the random effects implies a significant influence of other ecological covariate(s) on crop damage at the yearly and forest range levels that were not captured by the present study.

**Property damage:** A comparison of 220 property damage incidents recorded across the study area with the ecological covariates, including crop damage, showed that property damage by elephants decreased significantly in forest ranges with a higher grass biomass index, area of forest range, and extent of deciduous habitat, but it increased in forest ranges with a higher number of crop-damage incidents and greater extent of human settlement/cultivation (Table 3). The significant influence of seven covariates including the forest range and year from the random effects, along with the insignificant effect by nine other covariates, explained 82% of the variations in property-damage incidents.

**Human casualties:** Similarly, the comparison of 231 human casualties that took place across the study area with ecological covariates, including crop and property damage, showed that human casualties increased in forest ranges with more incidents of property damage, a higher ambient temperature, and larger forest range boundary perimeter, but decreased in forest ranges with a higher grass biomass index (Table 4). The significant influence from four covariates including the forest range and year from the random effect covariates, along with insignificant effects from 11 other covariates, explained 67% of the variation in human casualty incidents.

## 4. Discussion

The present study is the first study in Asia to empirically and comprehensively study human–elephant conflict with 26 ecological covariates including weather factors, grass dynamics factors, LULC elements, and elephant population and habitat factors. The study sheds new light, finding that grass dynamics covariates, viz. grass biomass index, a factor representing the quantity of grass and green grass, and a factor representing the quality of grass play a significant role in reducing human–elephant conflict.

### 4.1. Spatial Differences in HEC

The present findings showed that among the five forest divisions where the study assessed the HEC, the Coimbatore Forest Division [CFD] experienced the highest levels of HEC, except for the number of elephant electrocution incidents, which was highest in Hasanur. This trend is expected, as the CFD is under the territorial forest division. Forest division under this category is not exclusively meant for wildlife conservation in India. It also legally permits the local people to graze their cattle, and collect firewood and minor forest products. Such anthropogenic activities may contribute to a higher degree of habitat degradation in territorial forest divisions compared to the national parks and wildlife sanctuaries, which are part of the traditional Protected Areas network, and the tiger reserves, the latest concept included in this category. Among the Protected Areas, the national park and tiger reserves are exclusively meant for wildlife conservation; though the wildlife sanctuaries give high priority to wildlife conservation, a small part of them is open to human activities such as firewood collection and cattle grazing. In addition, the financial support available from the government sector for management is highest for the tiger reserves, followed by the national parks and wildlife sanctuaries. At the same time, the territorial division receives the lowest support, and thus, the protection status of the forest divisions is also the lowest. Therefore, the CFD, being a territorial division, experienced the highest level of HEC due to higher anthropogenic pressure. Further, its structural dimension as a linear stretch has also added to the HEC. Among the other four forest divisions, Sathiyamangalam and Hasanur, though together form the Sathiyamangalam Tiger Reserve (STR), still experienced a higher state of HEC compared to Mudumalai and Masinagudi that together form the Mudumalai Tiger Reserve (MTR). The STR was upgraded to its present Tiger Reserve status in 2013 and was a territorial forest division until 2012. The MTR was partly under a Wildlife Sanctuary status and partly under a National Park status until March 2007 and was declared a tiger reserve in April 2007. Thus, the higher HEC of the STR compared to the MTR could be attributed to the past status of the STR as a territorial division, with higher anthropogenic activities along with a large number of enclaves of settlements/cultivation areas. These findings are in a similar line to those of a recent study that assessed HEC across all forest divisions of Tamil Nadu [28], which showed that much of the highly affected HEC forest divisions are territorial divisions.

### 4.2. Temporal Differences in HEC

Among the four different consequences of HEC assessed, incidents of crop damage increased significantly from 2018 to 2023. This could be related to habitat loss or decreasing habitat quality or an increasing elephant population over the years. However, as the study demonstrated a negative relationship between elephant density and HEC incidents (i.e., crop and property damage and human casualties), the increasing trend in HEC in the NBR is likely due to decreasing habitat quality rather than an increasing elephant density. Habitat quality decreases due to a growing human population and its resultant rising anthropogenic pressure in the form cattle grazing and firewood and dung collection. Further, the increasing anthropogenic activities also increase weed proliferation [13] and forest fire frequency [38], which in turn decrease the availability of indigenous food plants for elephants and other herbivores. Nevertheless, considering the economic growth of the country and the fact that forested habitats in both the STR and MTR were upgraded into tiger reserves, which are known to better protect the habitats than territorial forest divisions, wildlife sanctuaries, and national parks in India, it is therefore the proliferation of alien invasive weeds like *Lantana camara* and *Chromolaena odorata*, which were reported in the study area [39], that is continuing the degradation of the habitat quality more than the direct anthropogenic pressure at present. The situation warrants the control or eradication of the alien invasive species and reducing the dependence of the neighboring community on natural habitats to enhance the habitat condition.

### 4.3. Drivers of HEC

#### 4.3.1. Negative Effect of Grass Dynamics Factor on HEC

The comparison of the ecological covariates with dependent factors such as crop and property damage and human casualties showed that all three dependent factors decreased with an increase in the grass biomass index, indicating that forest ranges with a higher grass biomass index experienced less crop and property damage and human casualties than those with a lower grass biomass index. Elephants are megaherbivores with poor digestion [40], and the quantity of their food is the crucial factor for their survival. Further, since elephants are highly adapted for grass feeding [41], elephants in the NBR predominantly feed on grass [13]. It is likely that in the forest ranges with a higher grass biomass index, elephants did not depend on crops as much, whereas in the forest ranges with a poor grass biomass index, elephants raided the crops field more frequently, resulting in crop damage showing negative relationship with the grass biomass index. Similarly, the finding that green grass was a negative predictor of crop damage indicates that crop fields in forest ranges with a higher availability of green grass or in the season with a higher availability of green grass experienced lower damage than those areas or times with a lower availability of green grass. Therefore, it is not only the quantity of grass, but the quality of grass that matters in determining the HEC level. Nevertheless, the finding that grass biomass was the most significant predictor in the model reveals the primary importance of quantity. This goes against the Forage Maturation Hypothesis [FMH], which predicts that herbivores prefer to consume plants at early phenological growth stages that are more digestible and hence maximize the net energy intake by foraging on soft-textured green plants [42]. While the forage quality declines with biomass [43] and forage intake declines at a low biomass, an intermediate forage biomass represents an optimal balance between forage quality and availability for many herbivores [44]. Elephants acting in contradiction to the FMH could partly be explained by the fact that elephants are large-bodied bulk feeders [14] that must continually process large volumes of plant material to meet their nutritional needs [45], which was speculated by [46], who showed that African elephants consistently selected foraging areas where plants were at later phenological growth stages than predicted by the FMH.

Similarly, the negative influence of the grass biomass index on property damage and human casualties could perhaps be attributed to the forest ranges having an inadequate grass biomass, which is the principal component of the diet of elephants, driving them to sustain themselves with the cultivated crops available in and around the areas. While doing so, they cause damage to properties around the crop fields and human casualties to the people who try to prevent them from damaging the crop fields and property. Therefore, the grass biomass index was a negative predictor of property damage and human casualties; at the same time, crop damage was the most significant driver of property damage, while property damage was the strongest determinant of human casualties. Therefore, the study clearly demonstrated that an inadequate grass biomass is the principal covariate driving the elephants to raid crop fields, which subsequently results in property damage and human casualties by elephants. With the increasing trend in HEC, both in Africa and Asia, studies on HEC have increased remarkably since 2000; no study empirically demonstrated that a lack of grass biomass or natural fodder is the key predictor of HEC. Nevertheless, there have been a few studies that speculated that elephants raid crops when the quality of their preferred food item, grass, begins to decline in their natural habitats [47], and crop-raiding is related to the palatability of plants [48,49]. One exclusive study that compared the causes and consequences of property damage in Asia and Africa [50] showed that property damage in South Luangwa, Africa, occurred toward the dry season, when staple crops have already been harvested and stored in the villages, indicating that African elephants seemingly followed the crops from the fields into the villages. Meanwhile, in Bardia, Nepal, and Manas, Northeastern India, the Asian elephants searched for food in properties all year round. The reasons for such a difference were attributed to the higher and easier availability of stored food in the Asian study areas, the stronger habituation of individual elephants, or the degradation level of the natural habitat, which was reported elsewhere [10].

#### 4.3.2. Negative Influence of Elephant Density and Positive Influence of Adult Male Proportion on HEC

The multivariate comparison of HEC with ecological covariates revealed that elephant population density negatively influences crop damage. In contrast, the proportion of adult male elephants in the population positively influences crop damage. Though not statistically significant, these covariates showed a similar trend with property damage and human casualties. The results indicate that the higher HEC observed in forest ranges is partly due to the higher proportion of adult male elephants in the population rather than the higher density of elephants. These findings are similar to those of earlier studies on Asian elephants in Southern India [51]. There is a sexual difference in crop raiding: females who live in matriarchal groups do not raid crops if their home is intact or not significantly degraded. On the other hand, males that leave maternal herds at puberty and lead solitary lives mostly learn crop-raiding from other males when they join other adult males, who are habitual crop-raiders, to form bachelor herds, even if their homes are by and large intact [52]. Therefore, the difference in crop-raiding behavior between males and female is that a male may raid crop fields even when its home is intact due learning, while females do not do so if their home is intact or not degraded. This means that female raiding could be attributed to the significant loss or degradation or fragmentation of their habitat, while male raiding is not entirely related to habitat loss or degradation. As the elephant population mostly consists of female herds rather than adult-bull herds, the negative trend of HEC with elephant density and the positive trend of HEC with the proportion of adult males indirectly indicate that there was no significant habitat loss in the study area.

#### 4.3.3. Positive Influence of Perimeter and Human Settlement/Cultivation Area and Negative Influence of Extent of Forest Range on HEC

The result of the present study also showed that crop damage and human casualties increased significantly with the perimeter of the forest ranges. It is likely that larger the perimeter of a forest range, the narrower the natural habitat, and thus the larger the non-forest elements, especially the human settlement/cultivation area, that are abutting the natural habitat and therefore, higher HEC is expected. The increasing perimeter of the forest range abutting human settlements would mean increased habitat degradation owing to higher anthropogenic pressures by more local people and cattle. Such a situation ultimately brings more people in contact with elephants and more elephants in contact with human settlement and cultivation areas, resulting in an increase in HEC with perimeter. The present findings support the earlier studies that showed a positive relationship between the perimeter of areas and crop-damage incidents in Africa [9] and Asia [7,51,53]. Similarly, the findings showed that HEC increased positively with the extent of human settlement/cultivation, and it was significant, especially in relation to property damage. A forest range with a higher extent of settlement/cultivation is likely to have higher fragmentation and/or a larger perimeter of human settlements/cultivation per unit of forest habitat, and thus, such ranges are expected to have higher HEC than those with a smaller perimeter. The finding is similar to that of [21], who showed that crop cover is one of the major drivers of HEC in the Eastern and Western Ghats of India, and [24], who showed increased crop depredation with an increase in the area of agriculture in north-eastern India. Further, our study showed an inverse relationship between HEC and the extent of forest area, indicating that forest ranges with a larger forest cover experienced less HEC than those with a smaller forest cover. Since HEC is a kind of resource overlap between humans and elephants, HEC is expected to decrease with an increase in natural habitats since the probability of contact per unit area decreases with increasing natural habitat.

#### 4.3.4. Influence of Tropical Dry-Thorn, Deciduous, and Semi-Evergreen/Evergreen Habitats on HEC

The study showed that the extent of deciduous and dry-thorn habitats by and large decreased the HEC, except for human casualties by the latter habitat, which showed a positive trend. In contrast, the semi-evergreen/evergreen habitat increased all HEC consequences, although not all were significant. The deciduous and dry-thorn habitats that represent secondary forests support a relatively higher grass biomass, unlike semi-evergreen/evergreen habitats, which characterize primary forests that contain a low grass biomass [18,51,54]. In support of the above statement, [32], who compared the NDVI with graminoid abundance and canopy cover, showed that the NDVI is negatively related to the total abundance of graminoids and positively related to canopy cover. As primary forests like evergreen habitats contain a denser and continuous canopy, and the secondary forests like deciduous or dry thorn forests consist of a lighter and discontinuous canopy, the grass biomass is lower in the former compared to latter habitats. Given the fact that elephants are predominantly grazers in the study area [18], and that the study also showed that grass biomass is a significant negative driver of HEC in the study area, the negative influence of secondary forests (deciduous and dry-thorn habitats) on HEC could be attributed to the higher availability of grass biomass in these habitats compared to semi-evergreen/evergreen habitats. A recent study on HEC that covered all the forest divisions of Tamil Nadu, Southern India, showed that higher HEC areas were associated with a higher NDVI and low temperatures [28], which supports our findings that HEC increases with the extent of primary forest, which is known for having a higher NDVI than secondary forest habitats. A possible explanation for the dry-thorn forests showing a positive trend with human casualties could be their existence at lower altitudes, much of which is abutting human settlement/cultivation areas on one side. Further, this habitat has the lowest visibility owing to dense bushes and is exposed to more anthropogenic pressure, resulting in more people encountering elephants at short distances in dry-thorn habitats than any other habitats.

### 4.4. Limitations and Scope for Future Research

The present study assessed 26 ecological variables, including grass phenology variables, and compared them with HEC. However, the nutrient composition of the elephant food plants, which includes both wild and cultivated crops, is known to influence HEC, especially the conflict arising from crop-raiding by elephants. The cultivated crops of cereal and millet provide significantly more protein, calcium, and sodium than the wild grasses of Sathiyamangalam, a part of the present study area, and thus elephant crop-raiding is attributed to optimal foraging [51]. Our study did not assess the effect of plant species nutrient composition on crop damage by elephants, which needs special attention in future studies. Further, the scale in which the present study assessed the drivers of HEC is small and a landscape-level assessment covering an entire population would shed a better light on HEC.

## 5. Conclusions

Our study comprehensively assessed 26 ecological covariates and compared them with the HEC incidents that took place between 2018 and 2023 across five forest divisions of the Nilgiri Biosphere Reserve. The results showed that HEC varied spatially, with territorial forest division experiencing the highest level and the crop-damage incidents increased over the years. The GLMM model used to identify the drivers of HEC is a new paradigm that showed that grass biomass is an essential driver of crop and property damage and human casualties. The findings reveal for the first time the importance of the adequacy of grass biomass in reducing HEC. The increase in HEC with the perimeter of forest ranges and the extent of human settlement/cultivation highlights the need for consolidating elephant habitats. Further, the negative relationship between HEC and elephant density and positive relationship with % of male adults indicates that the observed higher level of HEC in forest ranges is not due to the high-density population and is partly attributed to the higher % of males. The increasing HEC poses a significant challenge in protecting the people and elephants. Therefore, this study suggests following measures to reduce HEC in the study area: (i) reduce the anthropogenic pressure; (ii) control the weed proliferation, which will ultimately increase the grass biomass; (iii) consolidate elephant habitat including re-establishing elephant corridors and translocating enclave villages; and (iv) capture and domesticate the habitual crop-raiding males, which contribute to the major proportion of HEC in each area.

## Figures and Tables

**Figure 1 animals-14-03193-f001:**
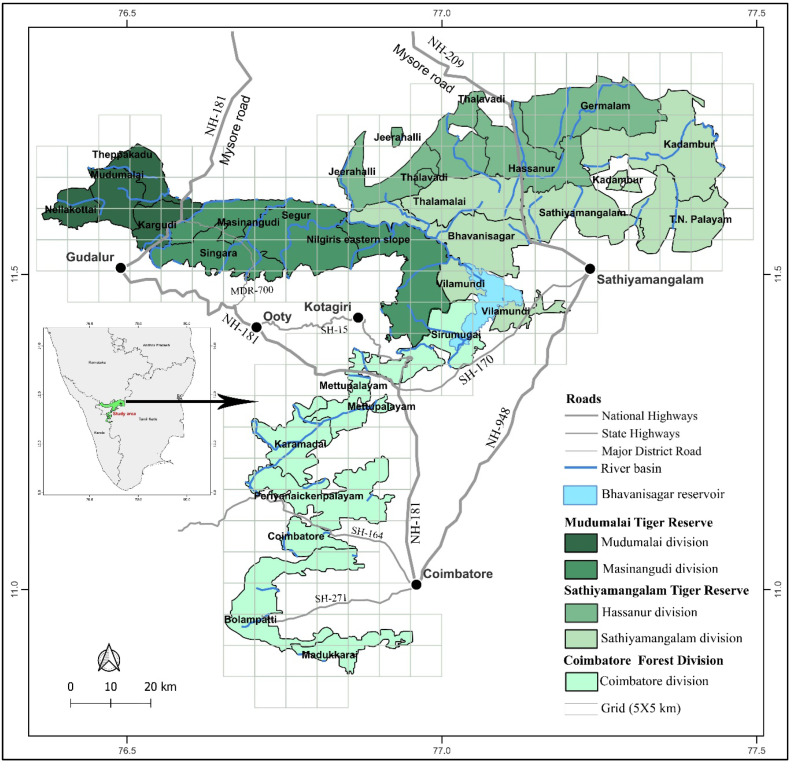
Map showing the layout of the twenty-five forest ranges and five forest divisions surveyed for human–elephant conflict in Nilgiri Biosphere Reserve, Western Ghats.

**Figure 2 animals-14-03193-f002:**
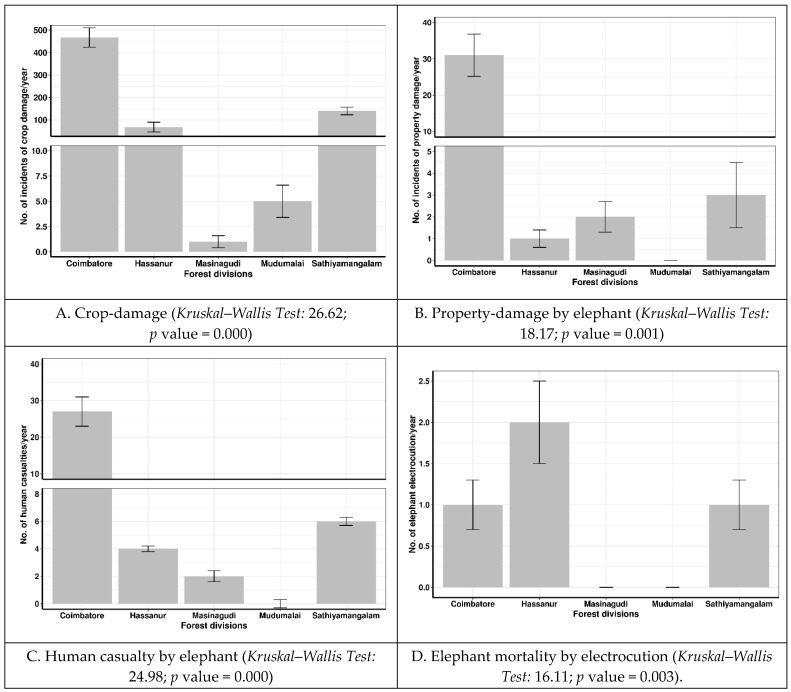
Spatial variation in human–elephant conflict recorded in Nilgiri Biosphere Reserve, Western Ghats, India.

**Figure 3 animals-14-03193-f003:**
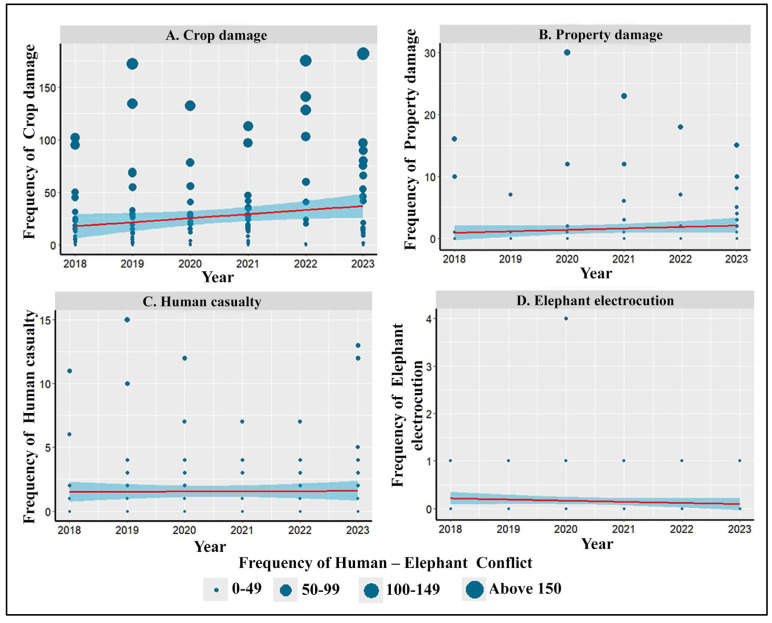
Temporal variation in human–elephant conflict recorded in the study area between 2018 and 2023. (**A**) Crop damage (R^2^ = 0.027; ß ± SE = 3.9 ± 1.92, *f =* 4.08, *p* = 0.045), (**B**) property damage (R^2^ = 0.009; ß ± SE = 0.2 ± 0.20, *f* = 1.34, *p* = 0.249), (**C**) human casualties (R^2^ = 0.007; ß ± SE = 0.02 ± 0.13, *f* = 0.02, *p* = 0.881), and (**D**) elephant electrocutions (R^2^ = 0.007; ß ± SE = −0.04 ± 0.04, *f* = 1.04, *p* = 0.294).

**Table 1 animals-14-03193-t001:** Details of ecological covariates assessed to compare with HEC in Nilgiri Biosphere Reserve, Southern India. The letters given in superscript ‘a’ refers to Methodology and ‘b’ refers to sampling Protocol given after each of them.

Category	Covariate	Sampling Location	^a^ Methodology and ^b^ Sampling Protocol
Weather	Rainfall (mm)	At the middle of each 25 km^2^ grid cell that falls within each forest range boundary	^a^ Data pertaining to rainfall, ambient temperature, and humidity were obtained from meteorological records of an online source: MERRA dataset (https://www.soda-pro.com/web-services/meteo-data/merra (accessed on 23 February 2024)). ^b^ All the three covariates were obtained for every grid cell within each forest range boundary. Monthly data on rainfall were averaged from multiple grid cells to arrive at the monthly rainfall and the same were totaled to obtain the annual rainfall for every forest range. The monthly ambient temperature and humidity data obtained from all the grid cells were averaged to arrive at the mean annual temperature and humidity for each forest range.
	2. Ambient temperature (°C)		
	3. Humidity (%)		
Grass dynamics	4. Grass height (cm)	8 quadrats, 1 m^2^ in size, per 25 km^2^ grid cell (*n* = 541)	^a^ Grass height was measured using a measuring scale from the ground level to the highest leaf blade bend, at five points: one each of the four corners and one at the center of the quadrat. ^b^ The study area was overlaid with 25 km^2^ grids and placed with 1 km lines transecting at alternating grid cells. At every 250 m interval along the transect, two 1 m^2^ quadrats were placed at a fifth of a meter on either side of the transect. Grass parameters were evaluated at bimonthly intervals and averaged for each grid cell first and later for the forest range to compare with the respective HEC data.
	5. Grass cover (%)		^a^ Grass cover % in area of quadrat occupied by grass, estimated visually. ^b^ Same as mentioned above for grass height.
	6. Grass biomass index		Estimated by multiplying the mean grass height with the mean grass cover obtained for each forest range.
	7. Soft-texture grass (%)		^a^ Assessed by visually quantifying the proportion of leaves with a soft texture, assuming 100% for all the leaves found within the quadrat. Soft-texture grasses were examined by crushing the leaves by hand; the proportion of leaves whose structure could be squashed into a ball for a given grass species in a quadrat was rated as a % rating. ^b^ Same as mentioned above for grass height.
	8. Hard-texture grass (%)		^a^ Assessed by visually quantifying the proportion of leaves with a hard texture, assuming 100% for all the leaves found within the quadrat. Hard-texture grasses were examined by crushing the leaves by hand; the proportion of leaves whose structure could not be squashed into a ball for a given grass species in a quadrat was rated as a % rating. ^b^ Same as mentioned above for grass height.
	9. Green grass (%)		^a^ Assessed by visually quantifying the proportion of leaves with green grass, assuming 100% for all the leaves found within the quadrat. ^b^ Same as mentioned above for grass height.
	10. Dry grass (%)		^a^ Assessed by visually quantifying the proportion of leaves with dry grass, assuming 100% for all the leaves found within the quadrat. ^b^ Same as mentioned as above for grass height.
	11. Reproductive phase (%)		^a^ Assessed by visually quantifying the proportion of grass with flowers, assuming 100% for all the grass found within the quadrat. ^b^ Same as mentioned as above for grass height.
Extent of various Land Use and Landcover (LULC)elements (km^2^)	12. Semi-evergreen/evergreen habitat	For each forest ranges (*n* = 25)	^a^ Land cover information was acquired from the Bhuvan website, which serves as a platform for accessing the satellite remote sensing data available to the public (https://bhuvan-app1.nrsc.gov.in/thematic/thematic/index.php (accessed on 23 February 2024)). Extent of each LULC element was extracted first for each grid cell and later for each forest range and compared with HEC intensity. ^b^ QGIS Version 3.34.2 was utilized to extract land cover data for each forest range within the study area.
	13. Tropical deciduous habitat		
	14. Tropical dry-thorn habitat		
	15. Human settlement/cultivation area		
	16. Water body area		
Elephant population and habitat	17. Elephant population size (total No. of individuals)	At each forest division (*n* = 5)	Obtained from reports of 2017 and 2023 Synchronized Elephant Censuses of Tamil Nadu (sample block count method), which are based on the Synchronized Elephant Census of Southern India by Project Elephant Govt. of India. Forest range areas were obtained from each Forest Range Office and cross-checked with the LUCL map used in this study.
	18. Elephant density (No./km^2^)		
	19. Adult male %		
	20. Forest range area (km^2^)		
	21. Forest ranges	*n* = 25	Each forest division consists of 4–7 forest ranges, which are the smaller administrative units. These are further divided into many beats, the smallest administrative units. The number of forest ranges found in each forest division were numbered as 1, 2, 3…25 and entered into the equation and compared with the respective HEC intensity.
	22. Perimeter of forest range (km)	For each forest ranges (*n* = 25)	The total length of the perimeter of each forest range was estimated using the study area map, and the perimeter of each forest range was compared with the respective HEC data. The perimeter of each forest range was derived from the study area map using QGIS Version 3.34.2, which was utilized to extract land cover data for each forest range within the study area.
Anthropogenic Pressure	23. Cattle grazing	At each grid cell, the 1 km transect was surveyed	^a^ Visually counted the number of cattle and firewood collectors found along the 1 km transect used for the grass dynamics covariate assessment and indexed as ‘0’ for absent, ‘1’ for a count of ≤5 individuals, and ‘2’ for a count of >5 individuals. Disturbances by people (both locals and tourists) were assessed considering whether public roads exist (an inter-state highway (3), state highway (2), road between two villages (1), or no road (0)). The numeric index for cattle grazing, firewood collection, and disturbance by people were averaged annually for each range and compared with the range’s annual HEC data. ^b^ Cattle and firewood collectors were counted at bimonthly intervals along the 1 km transect. Disturbance by local people and tourists was assessed for the entire grid cell.
	24. Firewood collection		
	25. Disturbance from local people and tourists		
Study period	26. Study years	2018–2023 (*n* = 6)	The trends in HEC consequences were compared over the study period from 2018 to 2023.

**Table 2 animals-14-03193-t002:** GLMM equation to identify the drivers of crop damage (*n* = 4073) by elephants in Nilgiri Biosphere Reserve.

Predictor	Estimate	Std. Error	*t*	*p*
(Intercept)	10.74577	3.659726	2.936223	0.003322
Soft-texture grass	−0.00294	0.032491	−0.09042	0.927952
**Grass biomass index**	**−0.00123**	**0.000356**	**−3.46159**	**0.000537**
Water body area	−0.1838	0.064179	−2.86388	0.004185
Ambient temperature (°C)	0.16123	0.085034	1.89602	0.057957
**Area of forest range (km^2^)**	**−0.006313**	**0.001408**	**−4.48258**	**7.37 × 10^−6^**
**Elephant density**	**−0.4656**	**0.235878**	**−1.97389**	**0.048395**
**Adult male elephants (%)**	**0.01784**	**0.003168**	**5.63229**	**1.78 × 10^−8^**
**Green grass (%)**	**−0.07719**	**0.034855**	**−2.21465**	**0.026784**
Semi-evergreen habitat (km^2^)	0.01954	0.010506	1.85977	0.062918
Human settlement/cultivation area (km^2^)	0.063305	0.036596	1.729842	0.083659
**Dry-thorn habitat (km^2^)**	**−0.22691**	**0.099469**	**−2.28122**	**0.022535**
**Perimeter of forest range (km)**	**0.023272**	**0.011825**	**1.968103**	**0.049056**
**Deciduous habitat (km^2^)**	**−0.01843**	**0.00708**	**−2.60325**	**0.009235**
Rainfall (mm)	0.000411	0.000497	0.827585	0.407906
**Random effect**				
Range code	sd__(Intercept)	0.745038		
Year	sd__(Intercept)	0.234092		
σ^2^	0.09			
τ_00 Range_	0.56			
_T00 Year_	0.05			
ICC	0.87			
N year	6			
N range code	25			
Observation	150			
Marginal R^2^	0.77			
Conditional R^2^	0.97			

Line in bold indicates statistically significant predictor.

**Table 3 animals-14-03193-t003:** GLMM equation to identify the drivers of property damage (*n* = 220) by elephants in Nilgiri Biosphere Reserve, Southern India.

Predictor	Estimate	Std. Error	Statistic	*p*
(Intercept)	3.0760674	10.0023261	0.308	0.758436
**Crop damage incidents**	**0.0114158**	**0.0034552**	**3.304**	**0.000953**
Soft-texture grass	0.0044821	0.0388401	0.115	0.908129
**Grass biomass index**	**−0.002504**	**0.0006686**	**−3.745**	**0.00018**
Water body area	−0.0014856	0.0233661	−0.064	0.949305
Ambient temperature (°C)	−0.3899258	0.3336823	−1.169	0.242583
**Area of forest range (km^2^)**	**−0.0103032**	**0.0031931**	**−3.227**	**0.001252**
Elephant density	−0.3621399	0.3252149	−1.114	0.265476
Adult male elephants (%)	0.1099256	0.0798625	1.376	0.168687
Semi-evergreen habitat (km^2^)	0.0005023	0.0161737	0.031	0.975226
**Human settlement/cultivation area (km^2^)**	**0.2496357**	**0.0726531**	**3.436**	**0.00059**
Dry-thorn habitat (km^2^)	−0.0892897	0.1692702	−0.527	0.597848
Perimeter of range (km)	0.0288189	0.0197564	1.459	0.144643
**Deciduous habitat (km^2^)**	**−0.0447218**	**0.0135625**	**−3.297**	**0.000976**
Rainfall (mm)	0.0027568	0.0015937	1.73	0.083668
**Random effect**				
Range code	sd__(Intercept)	0.26949		
Year	sd__(Intercept)	0.162542		
σ^2^	1.61			
τ_00 Range_	0.74			
_T00 Year_	0.28			
ICC	0.39			
N year	6			
N range code	25			
Observation	150			
Marginal R^2^	0.707			
Conditional R^2^	0.820			

Line in bold indicates statistically significant predictor.

**Table 4 animals-14-03193-t004:** GLMM equation to identify the drivers of human casualties (*n* = 231) due to elephants in Nilgiri Biosphere Reserve, Southern India.

Predictor	Estimate	Std. Error	Statistic	*p*
(Intercept)	−5.9211301	3.6488254	−1.623	0.1046
Crop damage incidents	0.002685	0.0033906	0.792	0.4284
**Property damage incidents**	**0.0358341**	**0.0183054**	**1.958**	**0.0503**
Soft-texture grass	−0.025348	0.0286553	−0.885	0.3764
**Grass biomass index**	**−0.0008996**	**0.0003929**	**−2.29**	**0.0220**
Water body area	−0.0409691	0.0685313	−0.598	0.5500
**Ambient temperature (°C)**	**0.253495**	**0.1019561**	**2.486**	**0.0129**
Area of forest range (km^2^)	−0.0006042	0.0016984	−0.356	0.722
Elephant density	−0.302591	0.2302102	−1.314	0.1887
Adult male elephants (%)	0.01284	0.0590548	0.217	0.8279
Semi-evergreen habitat (km^2^)	0.0171005	0.0119997	1.425	0.1541
Human settlement (km^2^)	0.0337928	0.0430646	0.785	0.4326
Dry-thorn habitat (km^2^)	0.0200018	0.121961	0.164	0.8697
**Perimeter of forest range (km)**	**0.025801**	**0.014073**	**1.833**	**0.0567**
Deciduous habitat (km^2^)	−0.0157012	0.0088759	−1.769	0.0769
Firewood collection	0.1044798	0.1560689	0.669	0.5032
**Random effect**				
Range code	sd__(Intercept)	0.189277		
Year	sd__(Intercept)	0.070658		
σ^2^	0.91			
τ_00 Range_	0.59			
_T00 Year_	0.01			
ICC	0.40			
N year	6			
N range code	25			
Observation	150			
Marginal R^2^	0.46			
Conditional R^2^	0.67			

Line in bold indicates statistically significant predictor.

## Data Availability

All data pertaining to the article are available within the manuscript. The corresponding author can be contacted for further inquiries.

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
