# Peer review of "Unveiling the Hidden Causes: Identifying the Drivers of Human–Elephant Conflict in Nilgiri Biosphere Reserve, Western Ghats, Southern India"

_animals, 2024, doi:10.3390/ani14223193_

Round 1
Reviewer 1 Report
Comments and Suggestions for Authors
A very good article. This is ready for publication with a few minor revisions.
A few minor edits: (1) Line 67 (firewood), (2) Line 362 (What is NDVI?)
If you argue that grass biomass is the essential driver of HEC, I would suggest that you include some photos of good (and not-so great biomass) for readers who are not familiar with the environment of India.
Map on P. 4 - Not clear. Where is the enlarged area located on the small map of India?
Comments on the Quality of English LanguageVery good. No revision will be necessary except for the above two minor points.
Author Response
|
Line number |
Comments |
Response |
Changes made |
Revised Line number |
|
Reviewer 1 |
||||
|
Line 67 |
firewood |
Agreed and suggestion incorporated |
Changed to ‘firewood’ and the term collection is suffixed with ‘s’, so that clarity improves. |
67 |
|
Line 362 |
What is NDVI? |
Agreed and Expansion is provided at the first mention of this term. |
Expanded as Normalized Difference Vegetation Index |
91 |
|
|
If you argue that grass biomass is the essential driver of HEC, I would suggest that you include some photos of good (and not-so great biomass) for readers who are not familiar with the environment of India |
Agreed and suggestion incorporated |
A couple of photographs showing grass growth are added now |
In supplementary (S-Fig. 1) |
Reviewer 2 Report
Comments and Suggestions for Authors
The content of this manuscript is very interesting and relevent. However the authors need to improve the grammar, sentence structure and punctuation throughout for clarity - there are far too many lengthy sentences that could be broken into two. There are some formatting issues e.g. Figure description must go beneath the figure and table description goes above the table. Just label Figure 1, not Figure 01.
Line 61-66: This is an example of a sentence that is too long and requires improved grammar.
Line 84: Should this be numbered '2' as Introduction is 1?
Line 85: replace 'is proposed for' with 'took place in'
Line 90: Change' wild animals round the year' to 'wild animals throughout the year'
Line 96: be consistent with use of '&' or 'and' in citations
Line 115: No need to repeat '2.2 methods' remove this and then have '2.2 Assess the intensity and driver.....' put in bold so consistent with previous sub-headings
Line 117: This sentence needs to be re-written for clarity e.g. '...determined the number of crops damaged, property damaged...' remove 'scrutinized to arrive at...'
Line 133: previously you italicized viz but in this section your haven't, keep consistent throughout.
Line 173: remove 'ing' from 'excepting'
Description for figure 2 must add 'Temporal variations in human-elephant conflict.....'
Line 268: remove 'the' from 'the human-elephant' and add 'conflict'.
Line 304: This sentence needs to be re-written for clarity and likely broken into two sentences. Also check citation font size is correct for Kodandapari et al.
Line 317: Remove 'Food is the basic requirement of any species' and add year for Benedict citation
Line 323: Instead of '...crop much...' change to 'crops as much'
Line 392: remove 's' from 'means'
Line 433: This should be in past tense 'Our study comprehensively assessed 26 ecological covariates and compared them......'
Line 435: Add full stop after 'Nilgiri Biosphere Reserve' forlled by 'We found that HEC....'
Comments on the Quality of English LanguageMany sentences I had to read numerous times to really understand the meaning. The content is there and it's very interesting, but I would suggest looking very closely at your sentence structure, tenses, and formatting to ensure that your point is clearly made as succinctly as possible.
Author Response
|
Line number |
Comments |
Response |
Changes made |
Revised Line number |
|
Reviewer 2 |
||||
|
Figure 1 |
There are some formatting issues e.g. Figure description must go beneath the figure and table description goes above the table. Just label Figure 1, not Figure 01. |
Agreed & suggestion incorporated. |
i.e. 01 changed to 1. |
Figure 1 160 |
|
Line 61-66 |
This is an example of a sentence that is too long and requires improved grammar.
|
Agreed and incorporated the changes. |
Changes in land use and landcover (LULC) elements, especially the conversion of private and revenue forests into agriculture/human settlement is expected to increase the HEC (Baskaran 2013, Sukumar 2016), as elephant clans show strong fidelity to their home due to hierarchy and its resultant spacing (Baskaran 1998, Baskaran et al. 2018). Further, the cultivated crops are more nutritive than wild-forages, as an optimal foraging strategy, adult bulls raid crop field (Sukumar 2016) |
61-65 |
|
Line 84 |
Should this be numbered '2' as Introduction is 1? |
Agreed and changed |
2. Methodology |
112 |
|
Line 85 |
replace 'is proposed for' with 'took place in' |
Agreed and changed. |
This study took place in Nilgiri Biosphere Reserve. |
114 |
|
Line 90 |
Change' wild animals round the year' to 'wild animals throughout the year' |
Agreed and changed ‘wild animals round the year’ to ‘wild animals throughout the year'. . |
…….., provide water sources to wild animals throughout the year |
119 |
|
Line 96 |
be consistent with use of '&' or 'and' in citations |
Agreed and replaced. |
(Champion & Seth 1968) |
125 |
|
Line 115 |
No need to repeat '2.2 methods' remove this and then have '2.2 Assess the intensity and driver.....' put in bold so consistent with previous sub-headings. |
Agreed and updated |
Deleted the '2.2 methods' added the same numbering 2.2 to Assess the intensity and driver...... Formatted the section number throughout uniformly. |
147 |
|
Line 117 |
This sentence needs to be re-written for clarity e.g. '...determined the number of crops damaged, property damaged...' remove 'scrutinized to arrive at...'
|
Agreed |
Rewritten as ‘Ex-gratia applications submitted by the affected people to the forest ranges belonging to five forest divisions were examined and arrived at the number of farmers / people experienced (i) crop damage, (ii) property damage, (iii) casualties by elephants and (iv) the number of elephants electrocuted in each forest range annually’. |
149-153 |
|
Line 133 |
Previously you italicized viz but in this section your haven't, keep consistent throughout. |
Agreed All the sub-headings are now formatted to maintain uniformity. |
|
|
|
Line 173 |
Remove 'ing' from 'excepting' |
Agreed |
In the GLMM equation, all the covariates, except the forest range and year, were treated as the fixed effect covariates, while, the forest range and year were treated as the random effect covariates. |
206 |
|
Figure 2 |
Description for figure 2 must add 'Temporal variations in human-elephant conflict.....' |
Agreed |
Figure 2 caption is added with ‘Temporal variations in elephant conflict.....’ |
246 |
|
Line 268 |
remove 'the' from 'the human-elephant' and add 'conflict'. |
Agreed - |
Remove word ‘the’ and added the word ‘conflict’ |
307 |
|
Line 304 |
This sentence needs to be re-written for clarity and likely broken into two sentences. Also check citation font size is correct for Kodandapari et al. |
Agreed and rewritten |
Habitat quality depletes due to growing human population and its resultant rising anthropogenic pressure in the form cattle-grazing, firewood, and dung collections. Further, the increasing anthropogenic activities also increase the weed proliferation (Baskaran et al. 2012) and forest fire frequency (Kodandapani et al. 2004), which in turn decrease the availability of indigenous food plants to elephants and other herbivores. |
348-352 |
|
Line 317 |
Remove 'Food is the basic requirement of any species' and add year for Benedict citation
|
Agreed. |
Elephants are megaherbivore with poor digestion (Benedict, 1936), and quantity of food is the crucial factor of their survival. |
367-369 |
|
Line 323 |
Instead of '...crop much...' change to 'crops as much' |
Agreed. |
….crops as much, whereas in the forest ranges with poor grass biomass index, elephants raided the crops-field more frequently resulting crop-damage showing negative relationship with grass-biomass index. |
372-374 |
|
Line 392 |
remove 's' from 'means' |
Though we removed it as per the suggestion of the reviewer, the language edit show that ‘s’ should be added and hence we retained it. |
This means that female raiding could be attributed to significant loss or degradation or fragmentation of habitat, while male raiding not entirely related to habitat loss or degradation |
435-436 |
|
Line 433 |
This should be in past tense 'Our study comprehensively assessed 26 ecological covariates and compared them......' |
Agreed and changed. However, conclusions drawn need not to be in past tense always. |
|
|
|
Line 435 |
Add full stop after 'Nilgiri Biosphere Reserve' forlled by 'We found that HEC....' |
Agreed and incorporated the changes |
…..across five forest divisions of Nilgiri Biosphere Reserve. The results showed that HEC varied spatially with territorial forest division experiencing the highest level and the crop-damage incidences increased over the years. |
499 |

Reviewer 3 Report
Comments and Suggestions for Authors
This study explores what environmental features are associated with different types of human-elephant incidents across five forest divisions of India. My main comment is that the somewhat disjointed presentation of the paper buries some of its most important results, which is the relationship between HEC types and quantitative measures of grass availability (and I can appreciate the trouble taken to measure these attributes!) in addition to other covariates. This is really the hidden gem – few studies have actually quantified this relationship so you should set up the expectations better in the introduction as motivating the study. It would also be advisable to broaden the perspective of the authors to provide more context for HEC generally; the introduction heavily cites the first author and little else, many other references are extremely dated. Many statements have no justifying references whatsoever. There have been decades of research on HEC, LULC change etc., both inside and outside India, so please make a reasonable effort to consult it as well as get a bit more up to date. A few references are provided at the end of this document as a starting point but are by no means meant to be exhaustive. In the discussion there are some interesting but odd and tangential arguments being made – such as explanations for why males and females might have different propensities to be involved in HEC. Not directly tied to the study – if these are speculative directions for future research, then say so explicitly (after making sure that someone hasn’t already made the argument elsewhere – if they have, cite that).
Several general comments below and more specific comments in the attached file:
Table 1 – It is difficult to understand what some of the variables actually are. Please explain clearly what is being quantified. See my in-text comments.
Throughout please use standard notations such as “per year” rather than terms like “annum” and use the notation in every reported quantity, rather than just the first set of parentheses.
Graph axes labels in the supplementary figure S-Fig. 01 are too small to be legible and figures are erratically placed in some sort of table. Please make all figures legible and fix their alignments. P-values cannot be zero even if very small; if please report as p< whatever the relevant threshold is.
Bar graphs do not need color scales that are redundant with what the axis itself is quantifying.
Why report both univariate and multivariate results when you’re using the same dependent variable and predictors in both cases? The latter does the job so unless I’m missing something you don’t need the simple regressions. Also it’s confusing that some results seem to contradict that of the multivariate analysis, unless stated incorrectly by mistake.
Some references in no particular order:
Tiller, L.N., Humle, T., Amin, R., Deere, N.J., Lago, B.O., Leader-Williams, N., Sinoni, F.K., Sitati, N., Walpole, M. and Smith, R.J., 2021. Changing seasonal, temporal and spatial crop-raiding trends over 15 years in a human-elephant conflict hotspot. Biological Conservation, 254, p.108941.
Webber, C.E., Sereivathana, T., Maltby, M.P. and Lee, P.C., 2011. Elephant crop-raiding and human–elephant conflict in Cambodia: crop selection and seasonal timings of raids. Oryx, 45(2), pp.243-251.
Munyao, M., Siljander, M., Johansson, T., Makokha, G. and Pellikka, P., 2020. Assessment of human–elephant conflicts in multifunctional landscapes of Taita Taveta County, Kenya. Global Ecology and Conservation, 24, p.e01382.
Thant, Z.M., May, R. and Røskaft, E., 2022. Human–elephant coexistence challenges in Myanmar: An analysis of fatal elephant attacks on humans and elephant mortality. Journal for Nature Conservation, 69, p.126260.
Gubbi, S., 2012. Patterns and correlates of human–elephant conflict around a south Indian reserve. Biological Conservation, 148(1), pp.88-95.
Buchholtz, E.K., McDaniels, M., McCulloch, G., Songhurst, A. and Stronza, A., 2023. A mixed‐methods assessment of human‐elephant conflict in the Western Okavango Panhandle, Botswana. People and Nature, 5(2), pp.557-571.
de Silva, S., Wu, T., Nyhus, P., Weaver, A., Thieme, A., Johnson, J., Wadey, J., Mossbrucker, A., Vu, T., Neang, T. and Chen, B.S., 2023. Land-use change is associated with multi-century loss of elephant ecosystems in Asia. Scientific reports, 13(1), p.5996.
Chiyo, P.I., Lee, P.C., Moss, C.J., Archie, E.A., Hollister-Smith, J.A. and Alberts, S.C., 2011. No risk, no gain: effects of crop raiding and genetic diversity on body size in male elephants. Behavioral Ecology, 22(3), pp.552-558.
de la Torre, J.A., Wong, E.P., Lechner, A.M., Zulaikha, N., Zawawi, A., Abdul‐Patah, P., Saaban, S., Goossens, B. and Campos‐Arceiz, A., 2021. There will be conflict–agricultural landscapes are prime, rather than marginal, habitats for Asian elephants. Animal Conservation, 24(5), pp.720-732.

Many grammatical and language errors throughout that interfere with reading and understanding. Please revise for clarity. Also check for typos (Land-Use and Land Cover is LULC not LUCL).
Author Response
|
Line number |
Comments |
Response |
Changes made |
Revised Line number |
|
Reviewer 3 |
||||
|
|
This study explores what environmental features are associated with different types of human-elephant incidents across five forest divisions of India. My main comment is that the somewhat disjointed presentation of the paper buries some of its most important results, which is the relationship between HEC types and quantitative measures of grass availability (and I can appreciate the trouble taken to measure these attributes!) in addition to other covariates. This is really the hidden gem – few studies have actually quantified this relationship so you should set up the expectations better in the introduction as motivating the study. It would also be advisable to broaden the perspective of the authors to provide more context for HEC generally; the introduction heavily cites the first author and little else, many other references are extremely dated. Many statements have no justifying references whatsoever. There have been decades of research on HEC, LULC change etc., both inside and outside India, so please make a reasonable effort to consult it as well as get a bit more up to date. A few references are provided at the end of this document as a starting point but are by no means meant to be exhaustive. In the discussion there are some interesting but odd and tangential arguments being made – such as explanations for why males and females might have different propensities to be involved in HEC. Not directly tied to the study – if these are speculative directions for future research, then say so explicitly (after making sure that someone hasn’t already made the argument elsewhere – if they have, cite that). |
Thank you for recognizing & appreciating the effort in evaluating the grass dynamics factor, which is laborious work.
Further, as pointed out by the Reviewer III, we have now set up the expectations better in the introduction by adding a paragraph, about the importance of food abundance in the context of large herbivores. Use of NDVI as an index of food availability by some studies and its disadvantage.
Thank you for the suggestion.
Self-citation been excluded especially in places >1 appeared. In addition, self-citation of unpublished reports also excluded.
Difference in crop raiding between the male & female is discussed in the context that why crop damage decreased with density of elephants, but increased with proportion of adult males in the population, despite the fact that adult males constitute <10% of the overall population.
As suspected by reviewer, the sexual difference in crop-raiding, is due to the male & female differences in sociality and probability of learning, which is not speculative, has been empirically demonstrated by Balasubramanian et al. 1995 and the same has been cited. |
|
74-97
69, 111, 414 and 486
418
432 |
|
|
Throughout, please use standard notations such as “per year” rather than terms like “annum” and the notation in every reported quantity, rather than just the first set of parentheses. |
Agreed and changed |
|
Figure 2 Line 247 |
|
|
Graph axes labels in the supplementary figure S-Fig. 01 are too small to be legible and figures are erratically placed in some sort of table. Please make all figures legible and fix their alignments. P-values cannot be zero even if very small; if please report as p< whatever the relevant threshold is. |
Agreed and changed. |
|
Fig. 2 in Supplementary file |
|
|
Bar graphs do not need color scales that are redundant with what the axis itself is quantifying. |
Agreed and changed. |
|
Figure 2 Line 247 |
|
|
Why report both univariate and multivariate results when you’re using the same dependent variable and predictors in both cases? The latter does the job so unless I’m missing something you don’t need the simple regressions. Also it’s confusing that some results seem to contradict that of the multivariate analysis, unless stated incorrectly by mistake. |
1) The Univariate is included into the MS, basically to show baseline data on HEC for the readers to view. Other-wise, we agree with the reviewer that univariate is not needed, when multivariate fulfils the requirement better.
2) We agree with the reviewer that an univariate result (relationship between crop-damage and % adult male in the population) is incorrectly (as negatively instead of positively) interpreted by mistake, which is now corrected.
Thank you for pointing out. |
2) But the HEC was higher in areas with higher adult male % in the population compared to areas of lower adult male % |
253
258
|
